# Effective Treatment of Diabetic Cardiomyopathy and Heart Failure with Reconstituted HDL (Milano) in Mice

**DOI:** 10.3390/ijms20061273

**Published:** 2019-03-13

**Authors:** Joseph Pierre Aboumsallem, Ilayaraja Muthuramu, Mudit Mishra, Herman Kempen, Bart De Geest

**Affiliations:** 1Centre for Molecular and Vascular Biology, Department of Cardiovascular Sciences, Catholic University of Leuven, 3000 Leuven, Belgium; josephpierre.aboumsallem@kuleuven.be (J.P.A.); illas1985@gmail.com (I.M.); mudit.mishra@kuleuven.be (M.M.); 2The Medicines Company (Schweiz) GmbH, CH-8001 Zürich, Switzerland; hermankempen@gmail.com

**Keywords:** type 2 diabetes mellitus, diabetic cardiomyopathy, obesity

## Abstract

The risk of heart failure (HF) is prominently increased in patients with type 2 diabetes mellitus. The objectives of this study were to establish a murine model of diabetic cardiomyopathy induced by feeding a high-sugar/high-fat (HSHF) diet and to evaluate the effect of reconstituted HDL_Milano_ administration on established HF in this model. The HSHF diet was initiated at the age of 12 weeks and continued for 16 weeks. To investigate the effect of reconstituted HDL_Milano_ on HF, eight intraperitoneal administrations of MDCO-216 (100 mg/kg protein concentration) or of an identical volume of control buffer were executed with a 48-h interval starting at the age of 28 weeks. The HSHF diet-induced obesity, hyperinsulinemia, and type 2 diabetes mellitus. Diabetic cardiomyopathy was present in HSHF diet mice as evidenced by cardiac hypertrophy, increased interstitial and perivascular fibrosis, and decreased myocardial capillary density. Pressure-volume loop analysis indicated the presence of both systolic and diastolic dysfunction and of decreased cardiac output in HSHF diet mice. Treatment with MDCO-216 reversed pathological remodelling and cardiac dysfunction and normalized wet lung weight, indicating effective treatment of HF. No effect of control buffer injection was observed. In conclusion, reconstituted HDL_Milano_ reverses HF in type 2 diabetic mice.

## 1. Introduction

Diabetic cardiomyopathy was first described in 1972 [1] and is characterized by the existence of ventricular dysfunction in the absence of other cardiac risk factors, such as coronary artery disease, hypertension, and significant valvular disease, in individuals with diabetes mellitus. In the first asymptomatic stage, diabetic cardiomyopathy includes a hidden subclinical period characterised by structural and functional abnormalities, including left ventricular hypertrophy and myocardial fibrosis, increased myocardial stiffness, and subclinical diastolic dysfunction. Subsequently, these abnormalities may evolve to heart failure with preserved ejection fraction (HFpEF) [2]. More pronounced systolic dysfunction may be accompanied by heart failure with reduced ejection fraction (HFrEF) [2]. 

Mechanisms leading to left ventricular impairment in type 2 diabetes are systemic changes. Not surprisingly, the right ventricle is also affected in patients with diabetic cardiomyopathy, as demonstrated by right ventricular remodelling and impaired systolic and diastolic function in men with type 2 diabetes, in a similar manner as changes in left ventricular dimension and left ventricular function [3]. Moreover, structural alterations occur predominantly in the right chambers of the heart during the early phase of experimental diabetes in rats [4]. 

Key metabolic abnormalities in type 2 diabetes mellitus are hyperglycemia, hyperinsulinemia, systemic insulin resistance, and impaired cardiac insulin metabolic signalling. Hyperglycemia, insulin resistance, and hyperinsulinemia induce metabolic alterations that lead to mitochondrial dysfunction, oxidative stress, advanced glycation end products (AGEs), impaired mitochondria Ca^2+^ handling, inflammation, activation of the renin–angiotensin–aldosterone system, endoplasmic reticulum stress, impaired myocardial microcirculation, and cardiomyocyte death [5]. A 1% reduction in haemoglobin A1c was associated with a 16% reduction of heart failure incidence in the UK Prospective Diabetes Study [6]. 

Type 2 diabetes mellitus impairs the capacity of the myocardium to use glucose as an energy source and fatty acid oxidation is increased in these subjects [7]. Increased subcellular vesicular recycling of cluster of differentiation 36 (CD36) from endosomes to the plasma membrane increases the rate of cellular uptake of free fatty acids in diabetic hearts [8]. Diabetic cardiomyopathy is associated with excess cardiac lipid accumulation [9]. Accumulation of lipid intermediates in the heart may lead to lipotoxicity characterised by cellular dysfunction, cardiomyocyte death, and deterioration of insulin resistance [9]. Interestingly, deficiency of CD36 rescues lipotoxic cardiomyopathy [10].

Epidemiological studies implicate added sugars in the development of the metabolic syndrome and type 2 diabetes mellitus [11,12,13]. In westernized cultures, the use of added sweeteners containing fructose (sucrose and high-fructose corn syrup) has increased by approximately 25% over the past three decades [14]. Fructose constitutes a particular toxic sugar challenge. It emerges that mice that cannot metabolize fructose are healthier when placed on carbohydrate-rich diets [15,16,17]. Fructose consumption may also impact the development of diabetic cardiomyopathy [5]. High-fructose diets induce cardiomyocyte autophagy, oxidative stress, and impaired insulin metabolic phosphatidylinositol 3-kinase (PI3K)/Akt/endothelial nitric oxide synthase (eNOS) signalling, and interstitial fibrosis [5].

Pleiotropic effects of high-density lipoproteins (HDL) including its anti-inflammatory, anti-oxidative, and anti-fibrotic properties may exert favourable effects on the myocardium [18,19,20,21]. We have previously shown that human *apolipoprotein (apo) A-I* gene transfer inhibits the development of diabetic cardiomyopathy in rats [22] and also improves diastolic function in hypercholesterolemic mice [23]. Furthermore, selective HDL-raising adeno-associated viral serotype 8-mediated human *apo A-I* gene transfer prevents heart failure induced by transverse aortic constriction in C57BL/6 low-density lipoprotein receptor-deficient mice [24]. However, an effect of HDL on established heart failure in a model of obesity and diabetes has never been investigated in an intervention study.

Apo A-I_Milano_ is an apo A-I mutant resulting from an arginine 173 to cysteine mutation and was discovered in 1980 in a family from Limone sul Garda in Northern Italy [25,26]. Heterozygous carriers of this mutant demonstrate apparent longevity [27] and are characterised by much less atherosclerosis than expected based on their plasma levels of HDL cholesterol (in the lowest 5th percentile (10–30 mg/dL)) [28]. MDCO-216 is a pharmaceutical product that contains reconstituted HDL comprising highly purified recombinant dimeric apoA-I_Milano_ complexed with 1-palmitoyl-2-oleoyl-sn-glycero-3-phosphatidylcholine (POPC) [29]. The objective of this study was to establish a robust murine model of diabetic cardiomyopathy induced by feeding a high-sugar/high-fat (HSHF) diet in C57BL/6N mice and to investigate the effect of intervention with reconstituted HDL_Milano_ (MDCO-216) on established heart failure in these diabetic mice.

## 2. Results

### 2.1. The HSHF Diet Induces Obesity and Type 2 Diabetes Mellitus in Female C57BL/6N Mice

The high-sugar/high fat (HSHF) diet was initiated at the age of 12 weeks. The time course of the body weight in standard chow (SC) diet and HSHF diet mice is shown in Figure 1A. Compared to SC diet mice, the body weight in HSHF diet mice was 1.16-fold (*p* < 0.0001) higher at 4 weeks, 1.30-fold higher (*p* < 0.0001) at 8 weeks, 1.39-fold (*p* < 0.0001) higher at 12 weeks (*p* < 0.0001), and 1.46-fold (*p* < 0.0001) higher at 16 weeks. The HSHF diet-induced diabetes mellitus (Figure 1B). Blood glucose levels in HSHF diet mice were 1.18-fold (*p* < 0.01) higher at 4 weeks, 1.26-fold (*p* < 0.0001) higher at 8 weeks, 1.32-fold (*p* < 0.0001) higher at 12 weeks, and 1.41-fold (*p* < 0.0001) higher at 16 weeks compared to SC diet mice. Plasma insulin (Figure 1C) and free fatty acids (Figure 1D) levels at the time of sacrifice were 3.56-fold (*p* < 0.0001) and 1.64-fold (*p* < 0.01) higher, respectively, in HSHF diet mice than in SC diet mice. Plasma adiponectin levels were reduced by 20.5% (*p* < 0.05) in HSHF diet mice compared to SC diet mice (Figure 1E). Taken together, the HSHF diet induces obesity, insulin resistance, and type 2 diabetes mellitus.

### 2.2. The HSHF Diet Induces Cardiac Hypertrophy and Pathological Remodelling in Female C57BL/6N Mice

Heart weight, left ventricular weight, and right ventricular weight were increased by 1.17-fold (*p* < 0.0001), by 1.18-fold (*p* < 0.0001), and by 1.25-fold (*p* < 0.05), respectively, in HSHF diet mice compared to SC diet mice (Table 1). Wet lung weight was 1.16-fold (*p* < 0.01) higher in HSHF diet mice than in SC diet mice, indicating the presence of heart failure. Kidney weight and spleen weight were increased by 1.06-fold (*p* < 0.05) and by 1.19-fold (*p* < 0.05), respectively, in HSHF diet mice compared to SC diet mice (Table 1).

Morphometric analysis corroborated left ventricular hypertrophy as indicated by increased left ventricular wall area (*p* < 0.0001), increased septal wall thickness (*p* < 0.001), and increased anterior wall thickness (*p* < 0.001) (Table 2). At the microscopic level, cardiomyocyte cross-sectional area was 1.48-fold (*p* < 0.0001) larger in HSHF diet mice than in SC diet mice (Table 2). Cardiomyocyte hypertrophy was paralleled by a decrease (*p* < 0.0001) of cardiomyocyte density (Table 2). Capillary density was 12.7% (*p* < 0.05) lower in HSHF diet mice than in SC diet mice. A pronounced increase of interstitial fibrosis (*p* < 0.0001) and perivascular fibrosis (*p* < 0.0001) was observed in HSHF diet mice. The 3-nitrotyrosine positive area was 3.51-fold (*p* < 0.0001) higher in HSHF diet mice than in SC diet mice, indicating increased nitro-oxidative stress. Taken together, the HSHF diet causes cardiac hypertrophy and pathological remodelling as evidenced by the reduced capillary density and the increased interstitial and perivascular fibrosis.

### 2.3. Cardiac Function Is Severely Compromised in HSHF Diet Mice

To evaluate cardiac function, pressure-volume data were generated in female C57BL/6N mice fed the SC diet or the HSHF diet using Millar Pressure-Volume (PV) Loop System (MPVS) and are summarized in Table 3. The HSHF diet-induced both systolic dysfunction and diastolic dysfunction. The preload recruitable stroke work (PRSW), which corresponds to the slope of the relationship between end-diastolic volume (EDV) and stroke work, and the end-systolic elastance (E_es_), which is the slope of the end-systolic pressure-volume relationship (ESPVR), are load-independent parameters of left ventricular contractility. PRSW was reduced by 39.9% (*p* < 0.001) in the HSHF diet mice compared to the SC diet mice. E_es_ was 51.0% (*p* < 0.0001) lower in HSHF diet mice than in SC diet mice. The effective arterial elastance (E_a_) describes the ability of the arterial system to accommodate pulsatile flow and was similar in both groups, reflecting a proportional reduction of end-systolic pressure (P_es_) and stroke volume in the HSHF diet group. Ventriculo-arterial coupling in HSHF diet mice was impaired as evidenced by the pronounced increase of the E_a_/E_es_ ratio (*p* < 0.0001). The slope of the end-diastolic pressure volume relationship (EDPVR) is a parameter reflecting the elastance or inverse of compliance of the left ventricular myocardium during the filling phase. The slope of EDPVR was significantly (*p* < 0.05) increased in HSHF diet mice compared to SC diet mice (Table 3). Isovolumetric relaxation was also impaired as evidenced by the significant decrease of the absolute value of dP/dt_min_ (*p* < 0.01) and the significant increase of the time constant of isovolumetric relaxation (tau) (*p* < 0.0001) in HSHF diet mice compared to SC diet mice (Table 3).

Stroke volume and cardiac output were 27.5% (*p* < 0.0001) and 26.8% (*p* < 0.001) lower, respectively, in HSHF diet mice than in SC diet mice. The peak filling rate (dV/dt_max_) (*p* < 0.01) and the absolute value of the peak emptying rate (dV/dt_min_) (*p* < 0.01) were significantly reduced in HSHF diet mice compared to SC diet mice (Table 3).

Taken together, the HSHF diet causes both systolic and diastolic dysfunction and induces impaired ventriculo-arterial coupling and a reduction of cardiac output.

### 2.4. Study Design and Metabolic Parameters in the MDCO-216 Intervention Study

The global study design of the intervention study evaluating the effect of reconstituted HDL_Milano_ on established heart failure in HSHF diet mice is illustrated in Figure 2. Endpoint parameters in reference SC diet mice and reference HSHF diet mice were determined at the age of 28 weeks. MDCO-216 SC diet and MDCO-216 HSHF diet intervention groups were treated with eight intraperitoneal administrations of 100 mg/kg (protein concentration) of MDCO-216 at an interval of 48 h each starting at the age of 28 weeks. Control buffer SC diet and control buffer HSHF diet mice were injected with an equal volume of buffer solution. Endpoint analyses in buffer mice and in MDCO-216 mice were performed at 30 weeks plus one day.

Lipid levels in SC diet mice and in HSHF diet are represented in Figure 3. Plasma cholesterol was increased by 2.60-fold (*p* < 0.0001), by 2.65-fold (*p* < 0.0001), and by 2.74-fold (*p* < 0.0001) in reference HSHF diet mice, buffer HSHF diet mice, and MDCO-216 HSHF diet mice, respectively, compared to respective SC groups (Figure 3A). These elevations corresponded to a 5.65-fold (*p* < 0.0001), a 5.91-fold (*p* < 0.0001), and a 6.74-fold (*p* < 0.0001) increase of non-HDL cholesterol in HSHF diet groups (Figure 3B) whereas the corresponding increases of HDL cholesterol were 1.53-fold (*p* < 0.0001), 1.49-fold (*p* < 0.05), and 1.46-fold (*p* < 0.01) (Figure 3C). Plasma triglyceride levels were not significantly different between different groups (Figure 3D).

The time course of body weight in the three SC diet groups and in the three HSHF diet groups is shown in Figure 4A. As expected, the weight difference was consistently similar between HSHF diet groups and respective SC diet groups. The HSHF diet-induced diabetes mellitus to a similar extent in all three HSHF diet groups (Figure 4B). A small but statistically significant (*p* < 0.05) reduction of blood glucose level at the time of sacrifice was observed in the MDCO-216 HSHF diet group compared to the buffer HSHF diet group. Plasma insulin levels were significantly increased in the HSHF diet groups compared to the respective SC diet groups (Figure 4C). Compared to reference SC diet mice and buffer SC diet, plasma insulin levels were reduced by 48.0% (*p* < 0.05) and by 52.0% (*p* < 0.001), respectively, in MDCO-216 SC diet mice. Plasma insulin level was 39.4% (*p* < 0.05) lower in MDCO-216 HSHF diet mice than in the reference HSHF diet mice (Figure 4C). Plasma free fatty acid levels were increased by 1.64-fold (*p* < 0.01), by 1.61-fold (*p* < 0.01), and by 1.36-fold (*p* < 0.05) in reference HSHF diet mice, buffer HSHF diet mice, and MDCO-216 HSHF diet mice, respectively, compared to respective SC diet groups (Figure 4D). Free fatty acid levels were 24.9% (*p* < 0.01) and 23.8% (*p* < 0.05) lower in MDCO-216 HSHF diet mice than in the reference HSHF diet mice and buffer HSHF diet mice, respectively (Figure 4D). Plasma adiponectin levels were significantly (*p* < 0.05) reduced in reference HSHF diet and in buffer HSHF diet mice compared to respective SC diet groups (Figure 4E). However, plasma adiponectin in the MDCO-216 HSHF diet group was not lower than in the MDCO-216 SC diet group and was 1.15-fold (*p* < 0.05) higher than in the reference HSHF diet group (Figure 4E). Taken together, blood glucose, plasma insulin, and plasma free fatty acids are significantly lower whereas plasma adiponectin is significantly higher in MDCO-216 HSHF diet mice than in the reference HSHF diet mice.

### 2.5. MDCO-216 Reverses Heart Failure and Partially Reverses Cardiac Hypertrophy and Pathological Remodelling in HSHF Diet Mice

Tissue and organ weights in buffer HSHF diet were similar than in the reference HSHF diet mice and the magnitude of increase compared to respective SC diet groups was also comparable for both HSHF groups (Table 4). Heart weight in MDCO-216 HSHF diet mice was 9.01% (*p* < 0.01) and 7.74 % (*p* < 0.05) lower than in reference HSHF diet mice and in buffer HSHF diet mice, respectively (Table 4). In addition, heart weight in MDCO-216 HSHF diet mice was not significantly different compared to MDCO-216 SC diet mice. Similar differences were observed for heart weight/tibia length ratio (Table 4). Left ventricular and right ventricular weights were not significantly higher in MDCO-216 HSHF diet mice than in MDCO-216 SC diet mice. Wet lung weight was increased by 1.16-fold (*p* < 0.01) and by 1.13-fold (*p* < 0.05) in reference HSHF diet mice and in buffer HSHF diet mice, respectively, compared to respective SC diet groups. However, no increase of wet lung weight was observed in the MDCO-216 HSHF diet group indicating the disappearance of lung congestion. Spleen weight in MDCO-216 HSHF diet mice was significantly reduced compared to reference HSHF diet mice (*p* < 0.01) and to buffer HSHF diet mice (*p* < 0.05) (Table 4).

At the histological level, capillary density in MDCO-216 HSHF diet mice was 1.15-fold (*p* < 0.05) and 1.11-fold (*p* < 0.05) higher than in reference HSHF diet mice and in buffer HSHF diet mice, respectively (Table 5). Moreover, perivascular fibrosis in MDCO-216 HSHF diet mice was decreased by 33.1% (*p* < 0.01) and by 40.6% (*p* < 0.001) compared to reference HSHF diet mice and buffer HSHF diet mice, respectively. The 3-nitrotyrosine positive was significantly (*p* < 0.01) lower in MDCO-216 HSHF diet mice than in the reference HSHF diet mice and in buffer HSHF diet mice. As expected, MDCO-216 did not result in any structural effect in SC diet mice (Table 4 and Table 5). All in all, intervention with MDCO-216 in HSHF diet induces a partial regression of cardiac hypertrophy and partially reverses pathological remodelling as evidenced by the increased capillary density and by the reduced perivascular fibrosis. Representative Sirius red-stained cross-sections of hearts of the three SC diet and of the three HSHF diet groups are shown in Figure 5. Representative photomicrographs illustrating laminin-stained cardiomyocytes, CD31-positive capillaries, Sirius red-stained collagen viewed under polarized light, and immunosections stained for 3-nitrotyrosine in the three SC diet groups and in the three HSHF diet groups are shown in Figure 6.

### 2.6. Prominent Restoration of Hemodynamic Function in HSHF Diet Mice Following Treatment with MDCO-216

An overview of hemodynamic data obtained using pressure-volume loop measurements in reference, buffer, and MDCO-216 SC diet and HSHF diet mice is provided in Table 6. MDCO-216 had no effect in SC diet mice that are characterised by a normal cardiac function. Intervention with buffer in HSHF diet mice did not have any beneficial effect on hemodynamic parameters and the magnitude of changes in reference HSHF diet mice and buffer HSHF diet mice compared to respective SC diet mice groups were highly similar. In contrast, intervention with MDCO-216 in HSHF diet mice reversed systolic (dP/dt_max_, PRSW, E_es_, dV/dt_min_) and diastolic abnormalities (dP/dt_min_, tau, dV/dt_max_) (Table 6). Stroke volume, cardiac output, and ventriculo-arterial coupling were normalized (Table 6). Taken together, treatment with MDCO-216 results in a prominent improvement of systolic and diastolic function in HSHF diet mice with a normalization of cardiac output and of ventriculo-arterial coupling.

### 2.7. MDCO-216 Significantly Improves Exercise Capacity in HSHF Diet Mice

Mice of the three SC diet groups and of the three HSHF diet groups were subjected to exercise treadmill testing to quantify exercise capacity. Lactate levels pre-exercise were not significantly different in the six groups (Figure 7A). The distance covered during exercise treadmill testing was reduced by 56.8% (*p* < 0.0001) and by 50.3% (*p* < 0.0001) in reference HSHF diet mice and in buffer HSHF diet mice, respectively, compared to respective SC diet groups (Figure 7B). The distance covered in MDCO-216 HSHF diet mice was 1.39-fold (*p* < 0.01) higher than in the reference HSHF diet mice and 1.32-fold (*p* < 0.05) higher than in buffer HSHF diet mice. Lactate post-exercise levels were significantly higher in reference HSHF diet (*p* < 0.001) and in buffer HSHF diet mice (*p* < 0.01) compared to respective SC diet groups (Figure 7C). Taken together, the HSHF diet reduces exercise capacity. Treatment with MDCO-216 significantly improves exercise capacity in HSHF diet mice.

### 2.8. MDCO-216 Induces a Pronounced Decrease of Plasma Tumor Necrosis Factor (TNF)-α Levels

To evaluate the impact of the HSHF diet and MDCO-216 intervention on systemic inflammation, plasma TNF-α levels were quantified (Figure 8). Plasma TNF-α levels were increased by 6.50-fold (*p* < 0.0001), by 6.52-fold (*p* < 0.0001), and by 3.25-fold (*p* < 0.0001) in reference HSHF diet mice, buffer HSHF diet mice, and MDCO-216 HSHF diet mice, respectively, compared to respective SC diet groups. Levels of TNF-α in MDCO-216 HSHF diet mice were decreased by 50.0% (*p* < 0.001) and by 49.3% (*p* < 0.01) compared to reference HSHF diet mice and buffer HSHF diet mice, respectively.

## 3. Discussion

The main findings of the current study are that (1) the HSHF diet induces obesity, type 2 diabetes mellitus, and diabetic cardiomyopathy; (2) the diabetic cardiomyopathy in this model is characterized by cardiac hypertrophy, capillary rarefaction, increased myocardial fibrosis, prominent cardiac dysfunction, and heart failure; (3) intervention with reconstituted HDL_Milano_ in HSHF diet mice reverses heart failure and partially reverses cardiac hypertrophy and pathological remodelling.

Structural and functional alterations and underlying mechanisms leading to diabetic cardiomyopathy in type 2 diabetes mellitus have been mostly investigated in db/db mice, ob/ob mice, Zucker diabetic fatty rats, and in diabetic patients [5]. Whereas ob/ob mice are leptin-deficient, db/db mice are leptin receptor-deficient [30]. In Zucker diabetic fatty rats, a mutation in the leptin receptor, OB-R, is associated with leptin resistance and obesity [31]. Due to single gene mutations that lead to the lack of action by the satiety factor leptin or its cognate receptor, these rodents spontaneously develop severe hyperphagia leading to obesity and manifest some characteristics of type 2 diabetes mellitus [30]. However, disease-causing genetic mutations in the leptin and leptin receptor are very rare in humans. Moreover, substantial differences exist between these animal models and human type 2 diabetes mellitus [30]. Considering that the use of added sweeteners containing fructose (sucrose and high-fructose corn syrup) may play a key potentiating role in the development of type 2 diabetes mellitus and associated diabetic cardiomyopathy in humans [5,11,12,13], an HSHF diet was applied in this study. Important components of this HSHF diet are fructose corn syrup-55 (17.5 weight percentage) and sucrose (17.5 weight percentage).

Previously, it has been demonstrated that short-term feeding of an HSHF diet for 8 weeks in female C57BL/6J starting from the age of 4 weeks induces insulin resistance and diastolic dysfunction as evidenced by echocardiographic analysis [32]. In the current study, the HSHF diet was initiated at the age of 12 weeks and was maintained for 16 weeks and induced type 2 diabetes mellitus in female C57BL/6N mice. The HSHF diet-induced diabetic cardiomyopathy in female C57BL/6N mice. Cardiomyopathy is defined by the European Society of Cardiology as a myocardial disorder in which the myocardium is both structurally and functionally abnormal, in the absence of coronary artery disease, arterial hypertension, valvular heart disease, and congenital heart disease sufficient to cause the observed abnormality of the heart muscle [33]. At the structural level, HSHF diet mice were characterized by left ventricular hypertrophy, cardiomyocyte hypertrophy, increased perivascular and interstitial fibrosis, and capillary rarefaction. At the functional level, both systolic and diastolic dysfunction were present. Systolic dysfunctions was evidenced by reduced left ventricular end-systolic elastance (E_es_) and decreased preload recruitable stroke work (PRSW), which are parameters of the systolic function that are load-independent. Multiple parameters in the model indicate the presence of diastolic dysfunction in HSHF mice: decreased left ventricular compliance (higher slope of the end-diastolic pressure volume relationship (EDPVR), reduced peak filling rate and impaired left ventricular isovolumetric relaxation. Taken together, the structural and functional data indicate that the HSHF mice represent a *bona fide* model of diabetic cardiomyopathy. These abnormalities resulted in a reduced cardiac output and an impaired ventriculo-arterial coupling as evidenced by the significantly increased E_a_/E_es_ ratio in HSHF diet mice. The E_a_/E_es_ ratio or ventriculo-arterial coupling ratio represents a measure of pump efficiency in expelling blood into the vasculature. The most favourable ventriculo-arterial coupling occurs when the E_a_/E_es_ ratio lies in the range 0.5–1.0. However, cardiac dysfunction does not automatically imply the presence of heart failure. The definition of clinical heart failure by the European Society of Cardiology is entirely based on clinical symptoms and signs [34]. The severe cardiac dysfunction in HSHF diet mice resulted in heart failure as evidenced by the increased wet lung weight and the severely reduced exercise capacity. Taken together, the HSHF diet mice constitute a model of diabetic cardiomyopathy and heart failure.

Intervention with reconstituted HDL_Milano_ in HSHF diet mice partially reversed cardiac hypertrophy and induced a partial regression of pathological remodelling in HSHF diet mice as evidenced by the increased capillary density and the decreased perivascular fibrosis. The anti-hypertrophic effects of HDL are consistent with prior in vitro and in vivo data. HDL has been shown to downregulate the angiotensin II type 1 receptor [35,36] and counteracts mechanical stress-induced autophagy and hypertrophy in cultured cardiomyocytes [37]. Moreover, continuous infusion of HDL inhibits cardiac hypertrophy in vivo [36,37], which may be mediated at least in part via downregulation of the angiotensin II type 1 receptor. We demonstrated that selective HDL-raising gene therapy also exerts anti-hypertrophic effects on the myocardium under conditions of pressure overload [24]. However, all these prior studies are not the equivalent of clinical intervention in patients with established heart failure since treatment is initiated before the onset of disease. Recently, regression of cardiac hypertrophy induced by administration of reconstituted HDL_Milano_ was demonstrated in a model of coconut oil-induced HFpEF [38]. This model was not characterized by diabetes mellitus or by hyperinsulinemia. The increase of capillary density and reduction of perivascular fibrosis reflects the pleiotropic effects of HDL. HDL exert multiple effects on the endothelium [39] and has potent effects on endothelial progenitor cells [40,41,42,43]. HDL reduces transforming growth factor-β 1-induced collagen deposition in murine fibroblasts [44] and decreases transforming growth factor-β1 in the myocardium [38]. Moreover, HDL has been shown to decrease transforming growth factor-β1-induced endothelial-mesenchymal transition in aortic endothelial cells in vitro [45]. Gene therapy with an E1E3E4-deleted human *apo A-I* gene transfer vectors reduced oxidative stress, inflammation, and myocardial fibrosis in a rat model of diabetic cardiomyopathy [22]. Adeno-associated viral serotype eight human *apo A-I* gene therapy strongly reduced myocardial fibrosis in a model of pressure overload-induced cardiomyopathy following transverse aortic constriction [24]. Finally, the regression of myocardial fibrosis induced by reconstituted HDL_Milano_ has recently been demonstrated in a murine model of HFpEF [38].

Intervention with MDCO-216 resulted in a prominent restoration of cardiac function in HSHF diet mice. Cardiac function at rest was similar in MDCO-216 HSHF diet mice compared to SC diet mice. Several mechanisms may contribute to increased cardiac function. First of all, it can be postulated that HDL has direct electrophysiological effects. HDL may regulate cholesterol distribution between the raft and non-raft membrane fractions [46]. Microdomain-specific localization of ion channels may affect their function [47]. Reconstituted HDL containing wild-type apo A-I shortened repolarization of isolated rabbit cardiomyocytes [48]. Moreover, infusion of reconstituted HDL shortened the heart-rate-corrected QT interval, which represents the duration of the ventricular electrical systole, on surface electrocardiograms in humans [48]. Direct effects of HDL have been shown in isolated cardiomyocytes in vitro as evidenced by activation of the transcription factor signal transducer and activator of transcription 3 (STAT3) via increased phosphorylation of extracellular signal-regulated kinases (ERK)1/2 [49] and by augmented phosphorylation of the pro-survival kinase Akt [22]. Interestingly, Scarb1^−/−^ mice that are deficient in scavenger receptor class B, type 1 are characterized by more pronounced cardiac hypertrophy, cardiac dysfunction, and heart failure under conditions of pressure overload [50]. HDL isolated from Scarb1^−/−^ mice that are deficient in scavenger receptor class B, type 1 is dysfunctional in activating Akt, ERK1/2, and STAT3 in isolated cardiomyocytes [50]. Finally, the effects of reconstituted HDL_Milano_ on the cardiac structure may also contribute to improved cardiac function. The increased capillary density and the regression of perivascular fibrosis may enhance cardiac function via improved myocardial microcirculation.

Autophagy is a cellular pathway for lysosomal degradation and recycling of long-lived proteins and organelles, which plays an important role in cardiac homeostasis [51,52]. Fructose-induced insulin resistance has been shown to increase autophagy, which may contribute to cardiac pathology [53]. Since HDL inhibits autophagy in cultured cardiomyocytes [37], this property may also contribute to the beneficial effects of MDCO-216 in this model of diabetic cardiomyopathy.

HDL modulate glucose metabolism [19]. HDL exert direct effects on adipose tissue, antagonizes lipolysis and enhances adiponectin expression [54]. Adiponectin is an adipokine that is downregulated in individuals with obesity-related disorders. HDL also improve peripheral glucose metabolism by phosphorylating AMP-activated protein kinase [55]. In this study, reconstituted HDL reduced free fatty acids concentrations, glucose levels, and insulin concentrations, and increased plasma adiponectin in HSHF diet mice. Adiponectin has insulin-sensitizing and anti-inflammatory effects [56]. This adipose-derived plasma protein also influences cardiac remodelling and suppresses pathological cardiac growth [57]. Taken together, the systemic effects of reconstituted HDL_Milano_ promote insulin sensitivity and may also contribute to the improvement of cardiac structure and function.

In contrast to adiponectin, levels of several other adipokines including tumour necrosis factor-α (TNF-α) increase in obesity. An inverse relationship exists between TNF-α and adiponectin is observed in humans [58,59]. TNF-α plasma levels were significantly decreased in MDCO-216 HSHF diet mice compared to the other HSHF diet groups. Obesity is characterised by chronic systemic inflammation originating from local immune responses in visceral adipose tissue. Infiltration of macrophages into adipose tissue follow the adipocyte-secretion of chemoattractants like TNF-α and free fatty acids [60]. Inflammation and heart failure are strongly interconnected and may reinforce each other in a mechanism of mutual causality [61]. TNF-α contributes to cardiac dysfunction and failure [62]. The cytokine hypothesis of heart failure postulates that heart failure progresses, at least in part, as a consequence of the harmful effects exerted by endogenous cytokine cascades on the heart and the peripheral circulation [63]. Whereas this hypothesis is essentially unproven, the increased anti-inflammatory potential of HDL and the reduction of TNF-α may have contributed to the beneficial effects of MDCO-216 on cardiac structure and function.

The improved cardiac function in MDCO-216 HSHF diet mice was paralleled by a significant amelioration of exercise capacity compared to the two other HSHF diet groups. The absence of complete normalization of exercise capacity compared to SC diet mice is likely due to the pronounced difference in weight, which requires the development of greater power during running on a 10° incline.

In conclusion, intervention with reconstituted HDL_Milano_ reverses diabetic cardiomyopathy and heart failure in a murine model of type 2 diabetes mellitus.

## 4. Materials and Methods

### 4.1. Reconstituted HDL_Milano_

MDCO-216 is a 1:1 by weight complex of recombinant dimeric apo A-I_Milano_ and 1-palmitoyl-2-oleoyl-sn-glycero-3-phosphatidylcholine (POPC) [38,64]. It was provided by The Medicines Company (Parsipanny, NJ, USA) as a solution in buffer containing mannitol 43.6 mM, sucrose 181 mM, NaH_2_PO_4_·2H_2_O 3.46 mM, and 8.43 mM Na_2_HPO_4_·7H_2_O.

The production of the recombinant protein in *Escherichia coli* and its purification have been described in detail by Caparon et al. [65]. Non-denaturing polyacrylamide gradient-gel electrophoresis demonstrated that the majority of the drug product displayed an apparent diameter of 8 nm and no free apoA-I_Milano_ was observed [66].

### 4.2. In Vivo Experiments and Study Design

All experimental procedures in animals were executed in accordance with protocols approved by the Ethical Committee Animal Experimentation of the Catholic University of Leuven (Approval number: P191/2015, approval date: 5 November 2015). C57BL/6N mice, originally purchased from Taconic (Ry, Denmark), were locally bred at the semi-specific pathogen-free facility of the Catholic University of Leuven at Gasthuisberg. The study design is illustrated in Figure 2. All experimental mice were of the female sex and were fed standard chow (SC) diet (Sniff Spezialdiäten GMBH, Soest, Germany) or were fed the pellet form of TestDiet 58Y1/5APC (London, WC1N3XX England, UK) starting at the age of 12 weeks. This experimental high-sugar/high fat (HSHF) diet was maintained for 16 weeks. The composition of TestDiet 58Y1/5APC (HSHF diet) is provided in Table 7. Metabolizable energy from standard chow is 13.5 MJ/kg (9 kJ% fat, 24 kJ% protein, 67 kJ% carbohydrates) whereas metabolizable energy from the HSHF diet is 19.5 MJ/kg (46.4 kJ% fat, 17.6 kJ% protein, 36.0 kJ% carbohydrates). Mono-and disaccharides in the HSHF diet consisted of high fructose corn syrup-55 (17.5%) and sucrose (17.5%). Reference SC diet mice and reference HSHF diet mice were analysed at the age of 28 weeks. MDCO-216 SC diet and HSHF diet intervention groups were treated with 8 intraperitoneal administrations of 100 mg/kg (protein concentration) of MDCO-216 at an interval of 48 h each starting at the age of 28 weeks. Control buffer SC diet and HSHF diet mice were injected with an equal volume of the buffer solution pH 7.4 containing mannitol 43.6 mM, sucrose 181 mM, NaH_2_PO_4_·2H_2_O 3.46 mM, and 8.43 mM Na_2_HPO_4_·7H_2_O (Figure 2). Endpoint analyses in the MDCO-216 and control buffer groups were performed at the age of 30 weeks plus one day, 24 h after the last injection. In the first experimental layer, mice were used for hemodynamic quantification and for histochemical and immunohistochemical analysis. The second experimental layer consisted of mice that did not undergo perfusion fixation and that were assigned for quantification of tissue and organ weights.

Group assignment at the start of the study was performed by randomisation. No single mouse died during the study. No mice were excluded from the analysis. Endpoint analyses were performed by investigators who were blinded to the group allocation of the mice. Unblinding of animal numbers corresponding to specific allocation groups was performed after completion of measurements and analyses.

### 4.3. In Vivo Hemodynamic Pressure-Volume Loop Measurements

Invasive hemodynamic measurements were performed before sacrifice following anaesthesia induced by intraperitoneal administration of 1.2 g/kg urethane (Sigma, St. Louis, MO, USA) [38]. Measurements were executed using Millar’s Mikro-Tip^®^ ultra-miniature pressure-volume (PV) loop catheter PVR-1035 (1.0 French polyimide catheter), the MPVS Ultra Single Segment pressure-volume unit, and a PowerLab 16/35 data acquisition system (ADInstruments Ltd., Oxford, UK) [38].

### 4.4. Quantification of Plasma Lipid Levels and Lipoprotein Cholesterol

Anticoagulation of blood obtained by puncture of the retro-orbital plexus was performed with 0.1 volume of 136 mmol/L trisodium citrate. Subsequently, plasma was immediately isolated by centrifugation at 1100 × *g* for 10 min and stored at −20 °C [64,67,68].

Plasma cholesterol and lipoprotein cholesterol levels were determined using a Cholesterol Quantification kit from Sigma-Aldrich (Sigma, St. Louis, MO, USA). HDL and non-HDL lipoproteins were separated by ultracentrifugation as described [69]. Plasma triglyceride concentration was quantified using the Triglyceride Quantification kit MAK266 (Sigma, St. Louis, MO, USA) according to the instructions of the manufacturer.

### 4.5. Quantification of Plasma Free Fatty Acid Levels

Plasma free fatty acids (FFA) levels were determined using Free Fatty Acids Quantification kit (Sigma-Aldrich, St. Louis, MO, USA) according to the instructions of the manufacturer.

### 4.6. Determination of Plasma Levels of Insulin, Adiponectin, and Tumor Necrosis Factor-α

Plasma insulin levels were measured using the Ultra-Sensitive Mouse Insulin enzyme-linked immunosorbent assay (ELISA) kit (Crystal Chem, Elk Grove Village, IL, USA). Murine plasma adiponectin was determined by ELISA according to the instructions of the manufacturer (Thermo Fisher Scientific, Vienna, Austria). Plasma tumour necrosis factor (TNF)-α levels were quantified using Mouse TNF-α Platinum ELISA (Thermo Fisher Scientific, Vienna, Austria).

### 4.7. Histological Analyses

Histological analyses were performed as described before [64,67,68]. After hemodynamic analyses, mice were perfused via the abdominal aorta with phosphate-buffered saline. Subsequently, hearts were arrested in diastole by KCl (100 μL; 0.1 mol/L), followed by perfusion fixation with 1% paraformaldehyde in phosphate-buffered saline. Thereafter, hearts were post-fixated overnight in 1% paraformaldehyde and embedded in paraffin. Cross-sections of 6 μm thickness at 130 μm spaced intervals were made extending from the apex to the basal part of the left ventricle. Comparative sections were analysed for all histological analyses by using the same slide numbers (1 to 40 from apex to base) and cross-section numbers (1–10).

To measure collagen content in the interstitium, Sirius Red staining was performed as described by Junqueira et al. [70]. Sirius Red polarisation microscopy on a Leica RBE microscope with KS300 software (Zeiss, Oberkochen, Germany) was applied to quantify thick tightly packed mature collagen fibres as orange-red birefringent and loosely packed less cross-linked and immature collagen fibres as yellow-green birefringent. Collagen positive area was normalised to the left ventricular wall area and was expressed as a percentage. Any perivascular fibrosis was excluded from this analysis. Perivascular fibrosis was quantified as the ratio of the fibrosis area surrounding the vessel to the total vessel area. Two mid-ventricular sections were studied per animal [67].

Cardiomyocyte hypertrophy was analysed on paraffin sections stained with rabbit anti-mouse laminin (Sigma; 1/50) by measuring the cardiomyocyte cross-sectional area (μm^2^) of at least 200 randomly selected cardiomyocytes in the left ventricular myocardium. Capillary density in the myocardium was determined on CD31 stained sections using rat anti-mouse CD31 antibodies (BD Biosciences, San Jose, CA, USA, 1/500). Relative vascularity was calculated as the ratio of capillary density to cardiomyocyte density divided by the cardiomyocyte cross-sectional area [71] and is expressed in µm^−2^. Two mid-ventricular cross-sections were analysed per mouse [72,73].

Immunostaining for 3-nitrotyrosine was performed with rabbit anti-nitrotyrosine antibodies (Merck Millipore, Overijse, Belgium; dilution 1/250).

### 4.8. Exercise Treadmill Testing

A motor-driven treadmill (Treadmill Simplex II, Columbus Instruments, Columbus, OH, USA) was applied to evaluate exercise capacity in mice [38,74]. C57BL/6N mice were familiarised with running on a motorized treadmill for one week. To quantify endurance capacity, mice started running on a 10° incline at an initial speed of 10 m/min, which was increased by 1 m/min every minute until the mouse resides on the stimulus plate (pulse grill) for ≥ 5 s. At this point, the mouse was immediately removed from the treadmill. The total exercise time was recorded as the elapsed time to exhaustion (min) and was then converted to distance (m), which is the end-point. Mice were subjected to tail snip before and after exercise tolerance test for lactate analysis (EKF diagnostics Company, Leipzig, Germany).

### 4.9. Statistical Analysis

At the end of the study, data of all surviving mice were included in the analysis. Investigators who performed endpoint analyses were blinded to group allocation. Unblinding of animal numbers corresponding to specific allocation groups was performed at the completion of measurements.

Statistical analysis was performed as outlined before [64,68]. Data are expressed as means ± standard error of the means (SEM). Minimally required sample size calculation (*n* = 13) for proving the effect of MDCO-216 on hemodynamic parameters in HSHF diet mice was based on a statistical power of 90%, a two-sided cut-off value of statistical significance of 0.05, a difference of main hemodynamic parameters at the population level of 20%, and a standard deviation at population level at 16% of the average of population means. Parameters between SC diet groups and respective HSHF diet groups were compared using Student’s *t*-test. When indicated, a logarithmic transformation or a non-parametric Mann–Whitney test was performed. The assumption of Gaussian distribution was tested using the Kolmogorov–Smirnov method. Parameters between the three SC diet groups or between the three HSHF diet groups were compared by one-way analysis of variance followed by Tukey’s multiple comparisons groups using GraphPad Instat (GraphPad Software, San Diego, CA, USA). When the assumption of sampling from populations with identical standard deviations was not met, a logarithmic transformation was performed. When the assumption of sampling from populations with Gaussian distributions was not met, a Kruskal-Wallis test was performed followed by Dunn’s multiple comparisons post-test. A two-sided *p*-value of less than 0.05 was considered statistically significant.

## Figures and Tables

**Figure 1 ijms-20-01273-f001:**
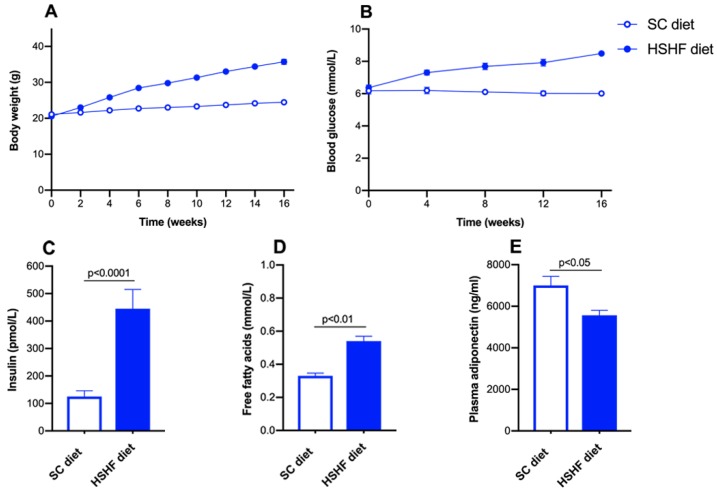
Time course of body weight (**A**) and blood glucose levels (**B**) in C57BL/6N mice fed the standard chow (SC) diet or the high-sugar/high-fat (HSHF) diet. Plasma insulin (**C**), free fatty acids (**D**), and adiponectin (**E**) levels at 16 weeks after the start of the diet. Week 0 in panels A and B corresponds to the age of 12 weeks, the start of the HSHF diet. All data represent mean ± SEM. (*n* = 15 for SC diet; *n* = 20 for HSHF diet).

**Figure 2 ijms-20-01273-f002:**
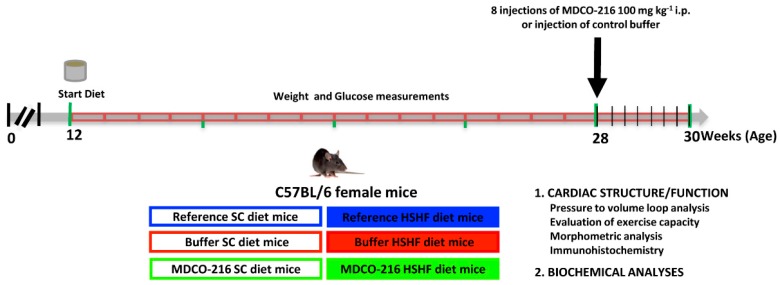
Schematic representation of the study design.

**Figure 3 ijms-20-01273-f003:**
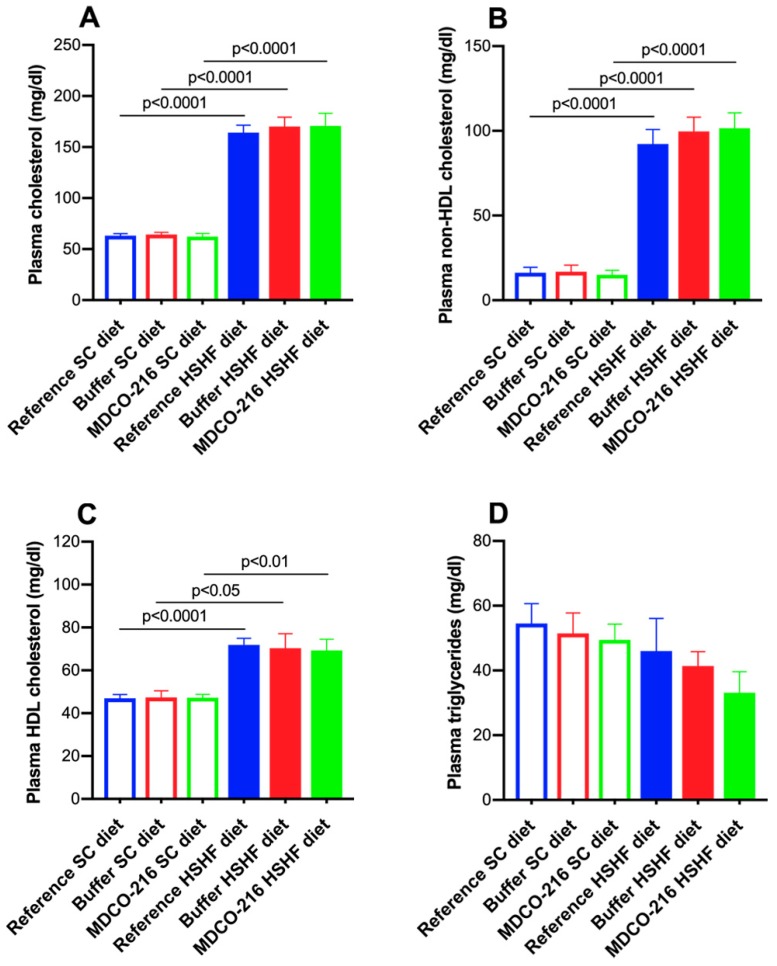
Total cholesterol (**A**), non-HDL cholesterol (**B**), and HDL cholesterol plasma levels (**C**) and plasma triglyceride levels (**D**) in female C57BL/6N mice at the time of sacrifice. All data represent means ± SEM (*n* = 8).

**Figure 4 ijms-20-01273-f004:**
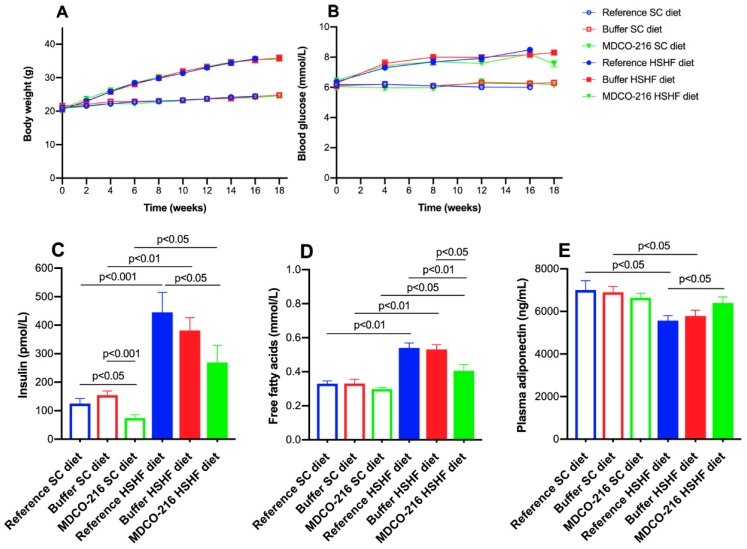
Time course of body weight (**A**) and blood glucose levels (**B**) in reference, buffer, and MDCO-216 C57BL/6N mice fed the SC diet or the HSHF diet. Plasma insulin (**C**), free fatty acids (**D**), and adiponectin (**E**) levels at the time of sacrifice. Week 0 in panels A and B corresponds to the age of 12 weeks, the start of the HSHF diet. All data represent mean ± SEM (*n* = 15 for SC diet groups; *n* = 20 for HSHF diet groups).

**Figure 5 ijms-20-01273-f005:**
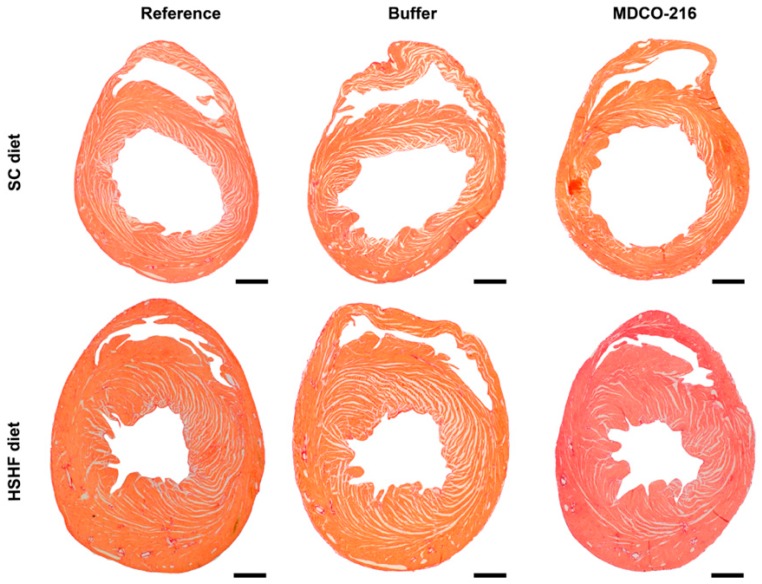
Representative Sirius-red stained heart cross-sections of reference, buffer, and MDCO-216 SC diet and HSHF diet mice. Scale bar represents 1 mm.

**Figure 6 ijms-20-01273-f006:**
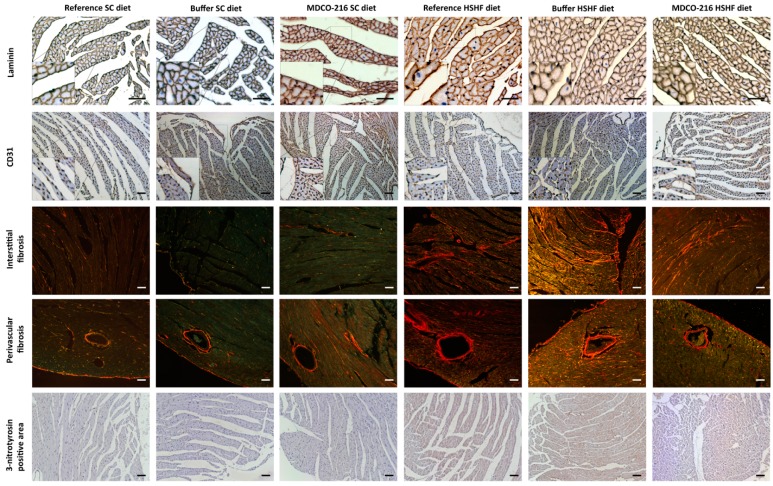
Immunohistochemical and histochemical analysis of the myocardium of reference, buffer, and MDCO-216 SC diet and HSHF diet mice. Representative photomicrographs show laminin-stained cardiomyocytes, CD31-positive capillaries, Sirius-red-stained collagen, and 3-nitrotyrosin positive area. Scale bar represents 50 µm.

**Figure 7 ijms-20-01273-f007:**
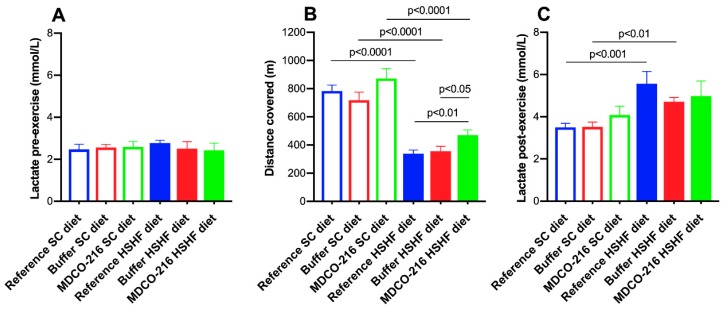
Lactate pre-exercise (**A**), distance covered (**B**), and lactate post-exercise (**C**) in reference, buffer, and MDCO-216 C57BL/6N mice fed the SC diet or the HSHF diet. All data represent means ± SEM (*n* = 10).

**Figure 8 ijms-20-01273-f008:**
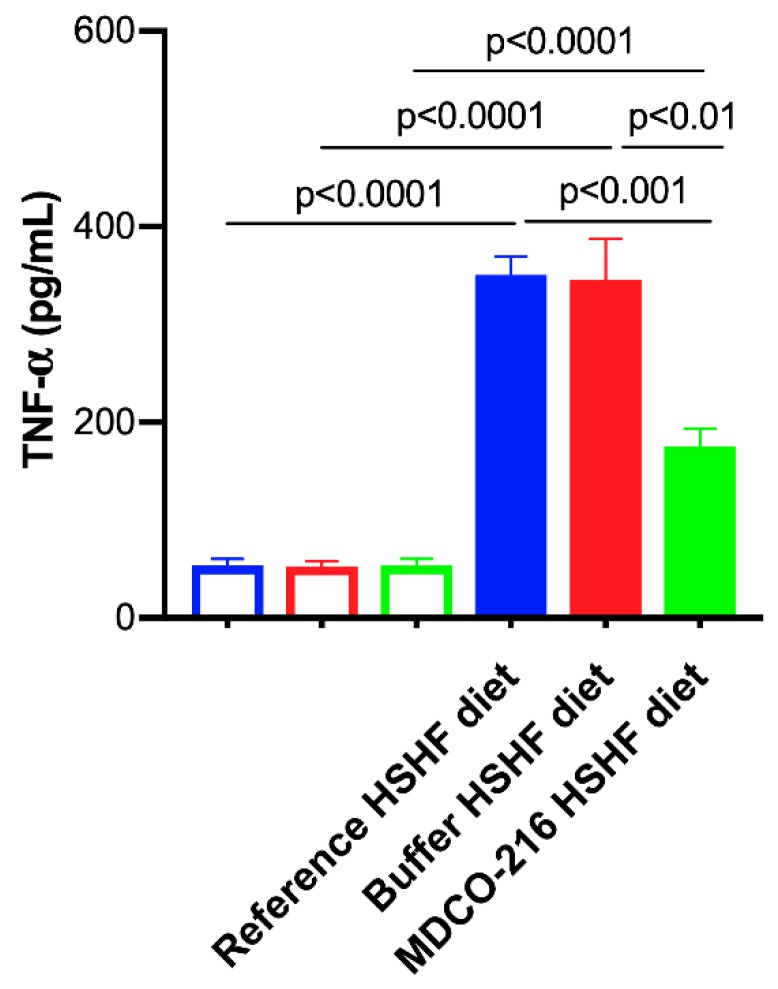
Plasma TNF-α concentration in the reference buffer, and MDCO-216 C57BL/6N mice fed the SC diet or the HSHF diet. All data represent means ± SEM (*n* = 10).

**Table 1 ijms-20-01273-t001:** Organ and tissue weights in female C57BL/6N mice fed the SC diet or the HSHF diet.

	SC Diet (*n* = 15)	HSHF Diet (*n* = 20)
Heart weight (mg)	119 ± 2	138 ± 3 ^§§§§^
Tibia length (mm)	17.6 ± 0.1	17.6 ± 0.1
Heart weight/tibia length (mg/mm)	6.73 ± 0.11	7.88 ± 0.16 ^§§§§^
Left ventricular weight (mg)	80.7 ± 1.5	95.1 ± 2.7 ^§§§§^
Right ventricular weight (mg)	18.6 ± 0.6	23.2 ± 2.2 ^§^
Lung weight (mg)	147 ± 5	171 ± 4 ^§§^
Liver weight (mg)	1090 ± 70	1070 ± 60
Kidney weight(mg)	306 ± 4	325 ± 6 ^§^
Spleen weight (mg)	71.3 ± 4.4	85.1 ± 2.7 ^§^

Both groups were sacrificed at the age of 28 weeks, which corresponds to 16 weeks after the start of the HSHF diet. All data are expressed as means ± SEM. ^§^: *p* < 0.05; ^§§^: *p* < 0.01; ^§§§§^: *p* < 0.0001 versus SC diet group.

**Table 2 ijms-20-01273-t002:** Morphometric and histological parameters of the left ventricular myocardium in C57BL/6N mice fed the SC diet or the HSHF diet.

	SC Diet (*n* = 20)	HSHF Diet (*n* = 21)
Left ventricular wall area (mm^2^)	7.97 ± 0.31	10.4 ± 0.4 ^§§§§^
Septal wall thickness (µm)	916 ± 41	1150 ± 50 ^§§§^
Anterior wall thickness (µm)	921 ± 33	1160 ± 40 ^§§§§^
Cardiomyocyte cross-sectional area (µm^2^)	181 ± 4	268 ± 10 ^§§§§^
Cardiomyocyte density (number/mm^2^)	4700 ± 190	3010 ± 130 ^§§§§^
Capillary density (number/mm^2^)	6070 ± 200	5290 ± 250 ^§^
Relative vascularity (µm^−2^)	0.00728 ± 0.00025	0.00682 ± 0.00038
Interstitial fibrosis (%)	2.34 ± 0.24	5.00 ± 0.29 ^§§§§^
Perivascular fibrosis (ratio)	0.227 ± 0.021	0.474 ± 0.038 ^§§§§^
3-nitrotyrosine positive area (%)	1.84 ± 0.14	6.47 ± 0.27 ^§§§§^

Histological and morphometric analyses in the reference groups were performed at the age of 28 weeks, which corresponds to 16 weeks after the start of the HSHF diet. All data are expressed as means ± SEM. ^§^: *p* < 0.05; ^§§§^: *p* < 0.001; ^§§§§^: *p* < 0.0001 versus SC diet group.

**Table 3 ijms-20-01273-t003:** Overview of hemodynamic data in C57BL/6N mice fed the SC diet or the HSHF diet.

	SC Diet (*n* = 12)	HSHF Diet (*n* = 15)
Heart rate (bpm)	606 ± 12	607 ± 13
P_max_ (mm Hg)	99.9 ± 1.5	79.6 ± 2.2 ^§§§§^
P_es_ (mm Hg)	97.7 ± 1.1	72.0 ± 2.1 ^§§§§^
dP/dt_max_ (mmHg/ms)	9.70 ± 0.80	7.73 ± 0.39 ^§^
PRSW (mm Hg)	87.5 ± 10.6	52.6 ± 4.5 ^§§§^
E_es_ (mmHg/µl)	8.25 ± 0.69	4.04 ± 0.34 ^§§§§^
P_min_ (mm Hg)	0.065 ± 0.820	1.88 ± 0.30 ^§^
P_ed_ (mm Hg)	2.06 ± 0.53	4.37 ± 0.31 ^§§§^
dP/dt_min_ (mmHg/ms)	−9.81 ± 0.69	−6.94 ± 0.21 ^§§^
Tau (ms)	5.52 ± 0.18	7.10 ± 0.23 ^§§§§^
Slope EDPVR (mmHg/µL)	0.316 ± 0.041	0.614 ± 0.143 ^§^
EDV (µL)	25.4 ± 0.9	23.5 ± 0.7
ESV (µL)	8.71 ± 1.00	11.4 ± 0.5 ^§^
Stroke volume (µL)	16.7 ± 0.7	12.1 ± 0.4 ^§§§§^
Ejection fraction (%)	66.3 ± 3.3	51.7 ± 1.3 ^§§^
Cardiac output (mL/min)	10.1 ± 0.5	7.39 ± 0.35 ^§§§^
Stroke work (mmHg.µL)	1330 ± 50	775 ± 39 ^§§§§^
dV/dt_max_ (µL/s)	859 ± 91	542 ± 29 ^§§^
dV/dt_min_ (µL/s)	−797 ± 61	−557 ± 47 ^§§^
E_a_ (mmHg/µL)	6.04 ± 0.39	6.02 ± 0.24
E_a_/E_es_	0.804 ± 0.109	1.60 ± 0.12 ^§§§§^

P_max_: maximum systolic pressure. P_es_: end-systolic pressure. dP/dt_max_: peak rate of isovolumetric contraction. PRSW: preload recruitable stroke work. E_es_: end-systolic elastance. P_min_: minimum diastolic pressure. P_ed_: end-diastolic pressure. dP/dt_min_: peak rate of isovolumetric relaxation. Tau: time constant of isovolumetric relaxation. EDPVR: end diastolic pressure-volume relationship. EDV: end-diastolic volume. ESV: end-systolic volume. dV/dt_max_: peak filling rate. dV/dt_min_: peak emptying rate. E_a_: arterial elastance. E_a_/E_es_: ventriculo-arterial coupling ratio. Hemodynamic measurements were performed at the age of 28 weeks, which corresponds to 16 weeks after the start of the HSHF die. All data are expressed as means ± SEM. ^§^: *p* < 0.05; ^§§^: *p* < 0.01; ^§§§^: *p* < 0.001; ^§§§§^: *p* < 0.0001 versus SC diet group.

**Table 4 ijms-20-01273-t004:** Organ and tissue weights in reference, buffer, and MDCO-216 C57BL/6N mice.

	Reference SC Diet (*n* = 15)	Buffer SC Diet (*n* = 15)	MDCO-216 SC Diet (*n* = 15)	Reference HSHF Diet (*n* = 20)	Buffer HSHF Diet (*n* = 20)	MDCO-216 HSHF Diet (*n* = 20)
Heart weight (mg)	119 ± 2	118 ± 3	120 ± 2	138 ± 3 ^§§§§^	136 ± 3 ^§§^	126 ± 3 ^°°*^
Tibia length (mm)	17.6 ± 0.1	17.6 ± 0.1	17.6 ± 0.1	17.6 ± 0.1	17.4 ± 0.1	17.5 ± 0.1
Heart weight/tibia length (mg/mm)	6.73 ± 0.11	6.74 ± 0.18	6.81 ± 0.13	7.88 ± 0.16 ^§§§§^	7.85 ± 0.20 ^§§^	7.20 ± 0.17 ^°°*^
Left ventricular weight (mg)	80.7 ± 1.5	81.8 ± 1.9	82.3 ± 2.3	95.1 ± 2.7 ^§§§§^	96.7 ± 3.1 ^§§§^	90.0 ± 3.2
Right ventricular weight (mg)	18.6 ± 0.6	17.3 ± 0.7	19.6 ± 1.0	23.2 ± 2.2 ^§^	22.6 ± 1.5 ^§§^	20.2 ± 0.9
Lung weight (mg)	147 ± 5	151 ± 4	153 ± 5	171 ± 4 ^§§^	170 ± 6 ^§^	154 ± 4 ^°°*^
Liver weight (mg)	1090 ± 70	971 ± 58	1050 ± 50	1070 ± 60	1070 ± 40	1070 ± 40
Kidney weight(mg)	306 ± 4	292 ± 8	291 ± 6	325 ± 6 ^§^	339 ± 9 ^§§^	324 ± 8 ^§^
Spleen weight (mg)	71.3 ± 4.4	69.3 ± 2.1	70.4 ± 2.2	85.1 ± 2.7 ^§^	93.9 ± 3.5 ^§§§§^	74.7 ± 2.2 ^°°***^

Reference groups were sacrificed at the age of 28 weeks, 16 weeks after the start of the HSHF diet. Eight intraperitoneal injections of reconstituted HDL_Milano_ (MDCO-216) (100 mg/kg) or of an equivalent volume of control buffer were executed with a 48-h interval starting at the age of 28 weeks. Mice in the buffer and MDCO-216 groups were sacrificed one day after the last injection. Kidney weight represents weight of both kidneys together. All data are expressed as means ± SEM. ^§^: *p* < 0.05; ^§§^: *p* < 0.01; ^§§§^: *p* < 0.001; ^§§§§^: *p* < 0.0001 versus respective SC diet group. ^°°^: *p* < 0.01 versus HSHF diet reference. ^*^: *p* < 0.05; ^***^: *p* < 0.001 versus HSHF diet buffer.

**Table 5 ijms-20-01273-t005:** Morphometric and histological parameters of the left ventricular myocardium in reference, buffer, and MDCO-216 C57BL/6N mice.

	Reference SC Diet (*n* = 20)	Buffer SC Diet (*n* = 15)	MDCO-216 SC Diet (*n* = 15)	Reference HSHF Diet (*n* = 21)	Buffer HSHF Diet (*n* = 17)	MDCO-216 HSHF Diet (*n* = 23)
Left ventricular wall area (mm^2^)	7.97 ± 0.31	8.48 ± 0.26	8.83 ± 0.38	10.4 ± 0.4 ^§§§§^	10.6 ± 0.4 ^§§§^	10.2 ± 0.4 ^§^
Septal wall thickness (µm)	916 ± 41	986 ± 46	985 ± 48	1150 ± 50 ^§§§^	1240 ± 40 ^§§§^	1120 ± 50
Anterior wall thickness (µm)	921 ± 33	973 ± 30	973 ± 47	1160 ± 40 ^§§§§^	1180 ± 30 ^§§§§^	1110 ± 30 ^§^
Cardiomyocyte cross-sectional area (µm^2^)	181 ± 4	187 ± 9	185 ± 10	268 ± 10 ^§§§§^	289 ± 8 ^§§§§^	251 ± 10 ^§§§§*^
Cardiomyocyte density (number/mm^2^)	4700 ± 190	4710 ± 210	4950 ± 290	3010 ± 130 ^§§§§^	3160 ± 90 ^§§§§^	3480 ± 130 ^°^
Capillary density (number/mm^2^)	6070 ± 200	6440 ± 250	6100 ± 260	5290 ± 250 ^§^	5520 ± 160 ^§§^	6110 ± 200 ^°*^
Relative vascularity (µm^−2^)	0.00728 ± 0.00025	0.00760 ± 0.00040	0.00698 ± 0.00032	0.00682 ± 0.00038	0.00615 ± 0.00022 ^§§^	0.00729 ± 0.00033 ^*^
Interstitial fibrosis (%)	2.34 ± 0.24	2.11 ± 0.10	2.01 ± 0.10	5.00 ± 0.29 ^§§§§^	4.28 ± 0.28 ^§§§§^	4.30 ± 0.23 ^§§§§^
Perivascular fibrosis (ratio)	0.227 ± 0.021	0.224 ± 0.013	0.202 ± 0.008	0.474 ± 0.038 ^§§§§^	0.534 ± 0.055 ^§§§§^	0.317 ± 0.029 ^§§°°***^
3-nitrotyrosine positive area (%)	1.84 ± 0.14	1.45 ± 0.14	1.74 ± 0.30	6.47 ± 0.27 ^§§§§^	6.38 ± 0.43 ^§§§§^	4.74 ± 0.36 ^§§§§°°**^

Histological and morphometric analyses in the reference groups were performed at the age of 28 weeks, 16 weeks after the start of the HSHF diet. Eight intraperitoneal injections of reconstituted HDL_Milano_ (MDCO-216) (100 mg/kg) or of an equivalent volume of control buffer were executed with a 48-h interval starting at the age of 28 weeks. Mice in the buffer and MDCO-216 groups were sacrificed one day after the last injection. All data are expressed as means ± SEM. ^§^: *p* < 0.05; ^§§^: *p* < 0.01; ^§§§^: *p* < 0.001; ^§§§§^: *p* < 0.0001 versus respective SC diet group. ^°^: *p* < 0.05; ^°°^: *p* < 0.01 versus HSHF diet reference. ^*^: *p* < 0.05; ^**^: *p* < 0.01; ^***^: *p* < 0.001 versus HSHF diet buffer.

**Table 6 ijms-20-01273-t006:** Overview of hemodynamic data in reference, buffer, and MDCO-216 C57BL/6N mice fed the SC diet or the HSHF diet.

	Reference SC Diet (*n* = 12)	Buffer SC Diet (*n* = 12)	MDCO-216 SC Diet (*n* = 12)	Reference HSHF Diet (*n* = 15)	Buffer HSHF Diet (*n* = 19)	MDCO-216 HSHF Diet (*n* = 22)
Heart rate (bpm)	606 ± 12	599 ± 19	610 ± 13	607 ± 13	578 ± 7	591 ± 13
P_max_ (mm Hg)	99.9 ± 1.5	99.4 ± 3.2	101 ± 1	79.6 ± 2.2 ^§§§§^	79.5 ± 2.4 ^§§§§^	93.1 ± 1.8 ^°°°§§***^
P_es_ (mm Hg)	97.7 ± 1.1	95.4 ± 3.2	95.5 ± 1.1	72.0 ± 2.1 ^§§§§^	72.5 ± 2.5 ^§§§§^	86.2 ± 2.2 ^§§°°°***^
dP/dt_max_ (mmHg/ms)	9.70 ± 0.80	10.6 ± 0.5	10.5 ± 0.4	7.73 ± 0.39 ^§^	7.34 ± 0.39 ^§§§§^	9.55 ± 0.46 ^°**^
PRSW (mm Hg)	87.5 ± 10.6	79.5 ± 6.5	76.6 ± 3.4	52.6 ± 4.5 ^§§§^	55.9 ± 4.3 ^§§^	73.0 ± 4.0 ^°°*^
E_es_ (mmHg/µL)	8.25 ± 0.69	8.24 ± 0.69	9.24 ± 0.47	4.04 ± 0.34 ^§§§§^	4.35 ± 0.33 ^§§§§^	7.11 ± 0.34 ^§§§°°°***^
P_min_ (mm Hg)	0.065 ± 0.820	1.01 ± 0.52	1.06 ± 0.58	1.88 ± 0.30 ^§^	2.00 ± 0.26	2.06 ± 0.25
P_ed_ (mm Hg)	2.06 ± 0.53	2.58 ± 0.44	2.58 ± 0.56	4.37 ± 0.31 ^§§§^	5.18 ± 0.33 ^§§§§^	4.04 ± 0.31 ^§*^
dP/dt_min_ (mmHg/ms)	−9.81 ± 0.69	−10.2 ± 0.8	−10.4 ± 0.4	−6.94 ± 0.21 ^§§^	−7.27 ± 0.42 ^§§^	−9.22 ± 0.31 ^§°°°***^
Tau (ms)	5.52 ± 0.18	5.52 ± 0.32	5.24 ± 0.21	7.10 ± 0.23 ^§§§§^	8.07 ± 0.28 ^§§§§^	5.48 ± 0.20 ^°°°***^
Slope EDPVR (mmHg/µL)	0.316 ± 0.041	0.371 ± 0.056	0.291 ± 0.056	0.614 ± 0.143 ^§^	0.454 ± 0.062	0.495 ± 0.064 ^§^
EDV (µL)	25.4 ± 0.9	24.9 ± 0.7	26.6 ± 1.4	23.5 ± 0.7	27.3 ± 1.1	23.8 ± 1.1 ^*^
ESV (µL)	8.71 ± 1.00	9.10 ± 0.40	9.76 ± 1.00	11.4 ± 0.5 ^§^	15.1 ± 1.0 ^§§§§°^	8.78 ± 0.74 ^°***^
Stroke volume (µL)	16.7 ± 0.7	15.8 ± 0.5	16.9 ± 1.1	12.1 ± 0.4 ^§§§§^	12.1 ± 0.8 ^§§^	15.0 ± 0.6 ^°°*^
Ejection fraction (%)	66.3 ± 3.3	63.5 ± 1.0	63.7 ± 3.0	51.7 ± 1.3 ^§§^	44.6 ± 2.5 ^§§§§^	64.1 ± 2.1 ^°°***^
Cardiac output (ml/min)	10.1 ± 0.5	9.46 ± 0.43	10.2 ± 0.6	7.39 ± 0.35 ^§§§^	7.01 ± 0.48 ^§§^	8.93 ± 0.44 ^°**^
Stroke work (mmHg.µL)	1330 ± 50	1260 ± 60	1360 ± 90	775 ± 39 ^§§§§^	765 ± 51 ^§§§§^	1120 ± 40 ^§°°°***^
dV/dt_max_ (µL/s)	859 ± 91	730 ± 38	768 ± 69	542 ± 29 ^§§^	587± 41 ^§^	688 ± 54 ^°^
dV/dt_min_ (µL/s)	−797 ± 61	−733 ± 42	−777 ± 68	−557 ± 47 ^§§^	−577 ± 44 ^§^	−732 ± 52 ^°*^
E_a_ (mmHg/µL)	6.04 ± 0.39	6.11 ± 0.29	5.87 ± 0.32	6.02 ± 0.24	6.66 ± 0.65	5.92 ± 0.26
E_a_/E_es_	0.804 ± 0.109	0.808 ± 0.081	0.699 ± 0.060	1.60 ± 0.12 ^§§§§^	1.56 ± 0.11 ^§§§§^	0.874 ± 0.061 ^°°°***^

P_max_: maximum systolic pressure. P_es_: end-systolic pressure. dP/dt_max_: peak rate of isovolumetric contraction. PRSW: preload recruitable stroke work. E_es_: end-systolic elastance. P_min_: minimum diastolic pressure. P_ed_: end-diastolic pressure. dP/dt_min_: peak rate of isovolumetric relaxation. Tau: time constant of isovolumetric relaxation. EDPVR: end diastolic pressure-volume relationship. EDV: end-diastolic volume. ESV: end-systolic volume. dV/dt_max_: peak filling rate. dV/dt_min_: peak emptying rate. E_a_: arterial elastance. E_a_/E_es_: ventriculo-arterial coupling ratio. Hemodynamic measurements in the reference groups were performed at the age of 28 weeks, 16 weeks after the start of the HSHF diet. Eight intraperitoneal injections of reconstituted HDL_Milano_ (MDCO-216) (100 mg/kg) or of an equivalent volume of control buffer were executed with a 48-h interval starting at the age of 28 weeks. Mice in the buffer and MDCO-216 groups were analysed one day after the last injection. All data are expressed as means ± SEM. ^§^: *p* < 0.05; ^§§^: *p* < 0.01; ^§§§^: *p* < 0.001; ^§§§§^: *p* < 0.0001 versus respective SC diet group. ^°^: *p* <0.05; ^°°^: *p* < 0.01; ^°°°^: *p* < 0.001 versus HSHF diet reference. ^*^: *p* < 0.05; ^**^: *p* < 0.01; ^***^: *p* < 0.001 versus HSHF diet buffer.

**Table 7 ijms-20-01273-t007:** Comparison of the composition of standard chow (SC) diet and TestDiet 58Y1/5APC (high-sucrose/high-fat (HSHF) diet).

Nutrient	SC Diet	HSHF Diet
Protein	19	20.5
Fat	3.3	24
Carbohydrates	41.3	41.8
Mono- and disaccharides	5.4	35
Fibre	24	5
Minerals	6.4	5.8

Data are expressed as a weight percentage.

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
