# Peer review of "Effective Treatment of Diabetic Cardiomyopathy and Heart Failure with Reconstituted HDL (Milano) in Mice"

_ijms, 2019, doi:10.3390/ijms20061273_

Round 1

Reviewer 1 Report

In this paper Joseph Pierre Aboumsallem and colleagues address a crucial topic in cardiovascular disease that is the increased risk of heart failure in patients with type 2 diabetes(T2D). The attention of the authors is focused on diet characterized by high amounts of sugar and saturated fat (HSHF ) which is very common in western countries and it has been associated with a higher risk for T2D. In this paper the authors aim to explore two main aspects: the first is the relation between HSHF diet-induced obesity, type 2 diabetes mellitus and diabetic cardiomyopathy; the second is the positive effect exerted by reconstituted HDLMilano treatment on HSHF diet-induced cardiac hypertrophy and pathological remodelling. To investigate the relationship between HSHF diet and cardiomyopathy the authors create a mouse model that is accurately described and characterized. All the parameters which are explored in the cardiomyopathy model are analyzed after HDLMilano treatments and the reversion of diabetic cardiomyopathy and heart failure by HDLMilano in this murine model of type 2 diabetes mellitus is clearly shown. An interesting aspects which is not addressed neither in the diabetic model nor in the treated model is the autophagy which is known to be affected by HDL levels. This might be an interesting point to assess.

Although many observations of this paper are not completely new, previous models or studies are extensively reported and well discussed. Overall this is an interesting paper which deserves to be published in the International Journal of Molecular Sciences.

Author Response

The authors thank the first reviewer for the constructive analysis of our manuscript.

In this paper Joseph Pierre Aboumsallem and colleagues address a crucial topic in cardiovascular disease that is the increased risk of heart failure in patients with type 2 diabetes(T2D). The attention of the authors is focused on diet characterized by high amounts of sugar and saturated fat (HSHF ) which is very common in western countries and it has been associated with a higher risk for T2D. In this paper the authors aim to explore two main aspects: the first is the relation between HSHF diet-induced obesity, type 2 diabetes mellitus and diabetic cardiomyopathy; the second is the positive effect exerted by reconstituted HDLMilano treatment on HSHF diet-induced cardiac hypertrophy and pathological remodelling. To investigate the relationship between HSHF diet and cardiomyopathy the authors create a mouse model that is accurately described and characterized. All the parameters which are explored in the cardiomyopathy model are analyzed after HDLMilano treatments and the reversion of diabetic cardiomyopathy and heart failure by HDLMilano in this murine model of type 2 diabetes mellitus is clearly shown. An interesting aspect, which is not addressed neither in the diabetic model nor in the treated model is the autophagy, which is known to be affected by HDL levels. This might be an interesting point to assess.

Although many observations of this paper are not completely new, previous models or studies are extensively reported and well discussed. Overall this is an interesting paper which deserves to be published in the International Journal of Molecular Sciences.

The following sentences have been added in the Discussion lines 412-416 in relation to HDL and autophagy:

‘Autophagy is a cellular pathway for lysosomal degradation and recycling of long-lived proteins and organelles, which plays an important role in cardiac homeostasis[1,2]. Fructose-induced insulin resistance has been shown to increase autophagy, which may contribute to cardiac pathology[3]. Since HDL inhibits autophagy in cultured cardiomyocytes[4], this property may also contribute to the beneficial effects of MDCO-216 in this model of diabetic cardiomyopathy’.

REFERENCES

1.         Terman, A.; Brunk, U.T. Autophagy in cardiac myocyte homeostasis, aging, and pathology. Cardiovasc Res 2005, 68, 355-365, doi:10.1016/j.cardiores.2005.08.014.

2.         Nemchenko, A.; Chiong, M.; Turer, A.; Lavandero, S.; Hill, J.A. Autophagy as a therapeutic target in cardiovascular disease. J Mol Cell Cardiol 2011, 51, 584-593, doi:10.1016/j.yjmcc.2011.06.010.

3.         Mellor, K.M.; Bell, J.R.; Young, M.J.; Ritchie, R.H.; Delbridge, L.M. Myocardial autophagy activation and suppressed survival signaling is associated with insulin resistance in fructose-fed mice. J Mol Cell Cardiol 2011, 50, 1035-1043, doi:10.1016/j.yjmcc.2011.03.002.

4.         Lin, L.; Liu, X.; Xu, J.; Weng, L.; Ren, J.; Ge, J.; Zou, Y. High-density lipoprotein inhibits mechanical stress-induced cardiomyocyte autophagy and cardiac hypertrophy through angiotensin II type 1 receptor-mediated PI3K/Akt pathway. J Cell Mol Med 2015, 19, 1929-1938, doi:10.1111/jcmm.12567.

Reviewer 2 Report

Dear authors, thank you for the work.

One minor concern. In the Introduction, you have postulated, that “In the first asymptomatic stage, diabetic cardiomyopathy includes a hidden subclinical period characterized by structural and functional abnormalities, including left ventricular hypertrophy and myocardial fibrosis, increased myocardial stiffness, and subclinical diastolic dysfunction”. However, the diabetic cardiomyopathy affects the right ventricle, as demonstrated by RV remodeling and impaired systolic and diastolic functions in men with type 2 diabetes, in a similar manner as changes in LV dimensions and functions [Diabetes Care 2013 Feb; 36(2): 457-462.]. Moreover, on early stages of diabetes, the most evident alterations are observed exactly in the right chambers of the heart [The Anatomical Record 298(2) DOI: 10.1002/ar.23052]. Probably, it’s better to rephrase the sentence.

Author Response

The authors thank reviewer 2 for the very positive comments on our manuscript.

Dear authors, thank you for the work.

One minor concern. In the Introduction, you have postulated, that “In the first asymptomatic stage, diabetic cardiomyopathy includes a hidden subclinical period characterized by structural and functional abnormalities, including left ventricular hypertrophy and myocardial fibrosis, increased myocardial stiffness, and subclinical diastolic dysfunction”. However, the diabetic cardiomyopathy affects the right ventricle, as demonstrated by RV remodeling and impaired systolic and diastolic functions in men with type 2 diabetes, in a similar manner as changes in LV dimensions and functions [Diabetes Care 2013 Feb; 36(2): 457-462.]. Moreover, on early stages of diabetes, the most evident alterations are observed exactly in the right chambers of the heart [The Anatomical Record 298(2) DOI: 10.1002/ar.23052]. Probably, it’s better to rephrase the sentence.

The authors thank the reviewer for pointing out that the right ventricle is also affected in diabetic cardiomyopathy. The following sentences have been added in the Introduction lines 41-46:

‘Mechanisms leading to left ventricular impairment in type 2 diabetes are systemic changes. Not surprisingly, the right ventricle is also affected in patients with diabetic cardiomyopathy, as demonstrated by right ventricular remodelling and impaired systolic and diastolic function in men with type 2 diabetes, in a similar manner as changes in left ventricular dimension and left ventricular function[1]. Moreover, structural alterations occur predominantly in the right chambers of the heart during the early phase of experimental diabetes in rats[2].’

REFERENCES

1.         Widya, R.L.; van der Meer, R.W.; Smit, J.W.; Rijzewijk, L.J.; Diamant, M.; Bax, J.J.; de Roos, A.; Lamb, H.J. Right ventricular involvement in diabetic cardiomyopathy. Diabetes Care 2013, 36, 457-462, doi:10.2337/dc12-0474.

2.         Danilova, I.G.; Sarapultsev, P.A.; Medvedeva, S.U.; Gette, I.F.; Bulavintceva, T.S.; Sarapultsev, A.P. Morphological restructuring of myocardium during the early phase of experimental diabetes mellitus. Anat Rec (Hoboken) 2015, 298, 396-407, doi:10.1002/ar.23052.